# GRAM2TOKEN: Enabling Run-time GPU-Native Grammar-Constrained Decoding for LLMs

Hantao Hua[1]    Jiming Su[2]    Hao Tang[1]    Yiping Yao[2]    Feng Zhu[2]

## Abstract

Grammar-constrained decoding enables large language models (LLMs) to reliably generate structured outputs such as JSON, SQL, and domain-specific programs. Existing systems often enforce constraints by executing byte-level parser logic inside the token-level decoding loop, introducing CPU-side control flow and CPU–GPU synchronization that become bottlenecks under continuous batching. We propose GRAM2TOKEN, a GPU-native framework that preprocesses deterministic byte-level grammar execution into token-level transitions before inference. GRAM2TOKEN aligns tokenizer byte sequences with grammar transitions through a trie and groups tokens with identical transition outcomes across preprocessed grammar states. These categories yield compact validity masks and transition tables, reducing run-time enforcement to category lookup, masking, and state update rather than parser-style byte traversal. Across four model families under schema-diverse continuous batching, GRAM2TOKEN achieves a geometric-mean throughput improvement of $1.38\times$ over the strongest baseline, with a maximum speedup of $1.85\times$, at the cost of additional preprocessing and time-to-first-token overhead. Break-even and grammar-complexity analyses show that this overhead is amortized by grammar reuse, longer outputs, and larger batches. These results show that token-level grammar preprocessing is an effective design point for high-throughput structured LLM serving. Code is available at https://github.com/Paradozile/Gram2Token.

[1]College of Computer Science and Technology, National University of Defense Technology [2]College of Systems Engineering, National University of Defense Technology. Correspondence to: Feng Zhu <zhufeng@nudt.edu.cn>.

*Proceedings of the $43^{rd}$ International Conference on Machine Learning*, Seoul, South Korea. PMLR 306, 2026. Copyright 2026 by the author(s).

## 1. Introduction

Large language models (LLMs) are increasingly deployed in applications that require structured outputs (Cai et al., 2023; Park et al., 2024), such as API function calls (Li et al., 2024), domain-specific language (DSL) (Li et al., 2025; Mündler et al., 2025), and JSON objects for parameter passing and information extraction (Cho et al., 2023). Unlike free-form text generation, these structured outputs must strictly adhere to grammar rules: a missing bracket in JSON, an ill-formed function call, or a malformed DSL command can render the output unusable (Chen et al., 2025; Geng et al., 2023; Park et al., 2025). However, enforcing these rules typically introduces a significant performance penalty. In large-scale or batch-oriented deployments, this computational overhead becomes particularly acute, often negating the throughput advantages of modern GPU hardware. This creates a critical need for methods that can maintain strict structural validity without sacrificing high-performance execution.

In practice, grammar-constrained decoding is commonly implemented by tracking grammar states during generation and masking invalid next tokens (Geng et al., 2023; Dong et al., 2025; Chen et al., 2025; Sun et al., 2025). However, formal grammars and their automata typically operate over characters or bytes, whereas LLMs generate variable-length tokenizer tokens. Existing systems therefore need to resolve byte-level grammar transitions inside the token-level decoding loop, often through parser-style state traversal, CPU-side control flow, and CPU–GPU synchronization. These operations are difficult to scale under continuous batching. Figure 1 illustrates this effect by reporting steady-state time-per-output-token (TPOT) on LLaMA3-8B (Touvron et al., 2023) as batch size increases: constrained decoding systems such as Pre[3] (Chen et al., 2025; Gong et al., 2025) and XGrammar (Dong et al., 2025) incur rapidly increasing per-token overhead at large batch sizes, limiting their suitability for throughput-oriented serving.

To address this mismatch, we propose GRAM2TOKEN, a GPU-native framework that moves grammar–token resolution out of the decoding loop. GRAM2TOKEN preprocesses deterministic byte-level grammar execution into token-level transition summaries by aligning tokenizer byte sequences with grammar transitions through a trie. It then groups

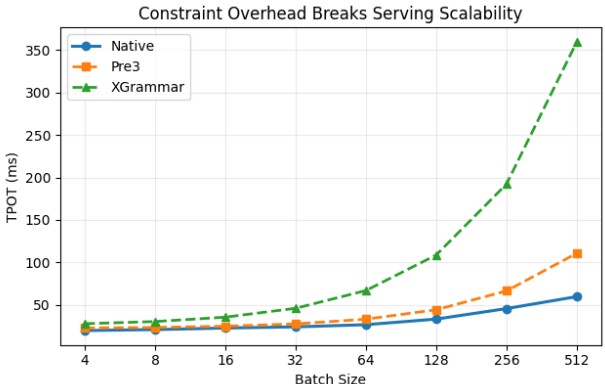

*Figure 1.* Steady-state per-token decoding overhead under constrained decoding on LLaMA3-8B. Existing parser-driven systems incur increasing TPOT at large batch sizes, while GRAM2TOKEN uses precomputed GPU-resident token/category-level tables for grammar handling. Preprocessing and time-to-first-token overheads are analyzed separately in Section 4.

tokens whose transition outcomes are identical across preprocessed grammar states, yielding compact GPU-resident validity masks and transition tables. At run time, each generated token directly indexes precomputed token–category tables to obtain its validity and successor grammar state, avoiding byte-by-byte grammar traversal on the critical path and making constrained decoding better match the GPU-parallel execution pattern of LLM inference.Building on this design, we make the following contributions:

We introduce a token-level preprocessing approach that aligns deterministic byte-level grammar execution with tokenizer outputs through a trie, replacing run-time byte traversal with precomputed token-level transitions.

We propose state-aware token categorization, which groups tokens with identical transition outcomes across preprocessed grammar states and substantially compacts GPU-resident validity masks and transition tables.

We design a GPU-native constrained decoding pipeline where each generated token directly indexes precomputed token–category tables for validity and successor-state lookup, avoiding run-time byte-by-byte grammar traversal and removing grammar-induced CPU–GPU synchronization from the steady-state critical path.

We evaluate GRAM2TOKEN on structured generation tasks under shared-grammar and mixed-schema batching. Across four model families under schema-diverse continuous batching, GRAM2TOKEN achieves a geometric-mean throughput improvement of $1.38\times$ over the strongest baseline, with a maximum speedup of $1.85\times$ and achieves larger gains over CPU-centric baselines. We further provide break-even and grammar-complexity analyses to characterize the preprocessing trade-off.

## 2. Preliminaries

### 2.1. Grammar-Constrained Decoding

Grammar-constrained decoding ensures that LLM outputs satisfy a predefined syntactic specification (Raspanti et al., 2025). Let $G = (N, \Sigma, P, S)$ denote a context-free grammar (CFG), where $N$ is the set of non-terminals, $\Sigma$ is the terminal alphabet, $P$ is the set of production rules, and $S$ is the start symbol. Given a generated token sequence $x = (x_1, \ldots, x_T)$, let $\text{str}(x)$ denote its realized output string. The goal of constrained decoding is to ensure $\text{str}(x) \in \mathcal{L}(G)$.

In practice, grammar enforcement is commonly implemented using an automaton that tracks the progress of grammar derivation during generation (Koo et al., 2024; Dong et al., 2025; Chen et al., 2025). For context-free specifications, this execution can be represented by a pushdown automaton (PDA), which maintains a control state and a stack. At decoding step $t$, we denote the grammar configuration as $s_t = (q_t, \sigma_t)$, where $q_t$ is the control state and $\sigma_t \in \Gamma^*$ is the stack content. The transition function $\delta$ updates this configuration as grammar terminals are consumed.

Since LLMs generate tokenizer tokens rather than grammar terminals, constrained decoding ultimately applies a validity mask over the token vocabulary $\mathcal{V}$. Let $\mathcal{A}(s_t) \subseteq \mathcal{V}$ denote the set of tokens whose realized byte sequence can be validly consumed from grammar configuration $s_t$. Given the model distribution $p(v \mid x_{<t})$ over tokens, constrained decoding applies

$$\tilde{p}(v \mid x_{<t}) \propto \begin{cases} p(v \mid x_{<t}) & v \in \mathcal{A}(s_t) \\ 0 & \text{otherwise} \end{cases}$$

so that only grammar-valid token continuations can be sampled.

For grammars with deterministic execution, grammar transitions can be represented by a deterministic pushdown automaton (DPDA), enabling unambiguous state updates without backtracking. Following prior work such as Pre[3] (Chen et al., 2025), we focus on target grammars that admit such a DPDA-style execution model (Valiant, 1973). Under this assumption, consuming a valid byte sequence from a grammar state yields a unique successor grammar state, which allows the effect of a tokenizer token to be summarized as a token-level transition.

Importantly, the grammar automaton above operates over grammar terminals, which are typically character- or byte-level symbols rather than tokenizer tokens. Constructing an automaton directly over the full tokenizer vocabulary would be prohibitively large, since modern vocabularies contain tens of thousands of variable-length tokens. Therefore, constrained decoding systems must determine, for each

grammar state and tokenizer token, whether consuming the token's byte sequence is valid and what successor grammar state it induces. The next subsection introduces the token-to-byte representation used to formalize this mismatch.

## 2.2. Tokenization and Token Representations

LLMs generate text as tokenizer tokens rather than raw bytes. Modern tokenizers, such as byte-pair encoding (BPE) (Berglund & van der Merwe, 2023) and Sentence-Piece (Kudo & Richardson, 2018), define a vocabulary $\mathcal{V}$ where each token $v \in \mathcal{V}$ corresponds to a variable-length byte sequence. We write

$$\phi : \mathcal{V} \rightarrow \Sigma^+$$

for the mapping from a token to its realized byte sequence, where $\Sigma$ denotes the byte alphabet.

During decoding, emitting a token $v$ conceptually consumes the entire byte sequence $\phi(v)$ under the grammar automaton. Thus, from a grammar state $s$, token validity is determined by whether the byte-level execution on $\phi(v)$ is valid, and the successor grammar state is determined by the resulting automaton configuration. A runtime parser-based system must resolve this byte-level effect for candidate tokens during decoding, which is the source of the byte–token mismatch discussed above.

Tokenizer vocabularies can be represented as tries (De La Briandais, 1959; Dori & Landau, 2006). In such a trie, each edge is labeled by a byte, and each root-to-leaf path corresponds to the byte sequence of a token. Shared byte prefixes among tokens are represented by shared internal nodes. This structure is useful for grammar-constrained decoding because it allows the byte-level effect of many tokens to be propagated jointly over shared prefixes, rather than simulated independently for every token.

Figure 2 illustrates a tokenizer trie, where tokens with common byte prefixes share the same traversal before diverging at later bytes.

## 2.3. Runtime Grammar States

During auto-regressive inference, the model maintains an inference state for the generated prefix and produces a distribution over the next token (Vaswani et al., 2017). When grammar constraints are applied, decoding additionally maintains a runtime grammar state that summarizes the effect of the generated output on the underlying grammar execution. After a token is emitted, its byte sequence updates this grammar state, and the updated state determines which tokens are valid at the next step.

Following Pre[3] (Chen et al., 2025), we assume an LR(1)-style, byte-consuming deterministic grammar execution

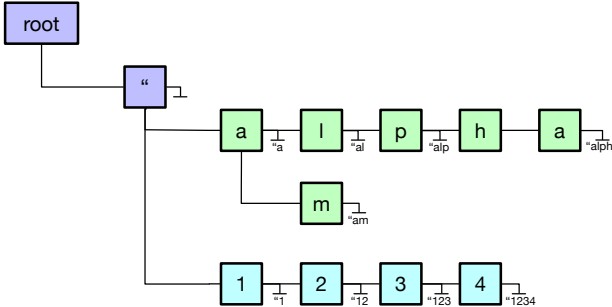

*Figure 2.* Illustration of a tokenizer trie. Each root-to-leaf path corresponds to the byte sequence of a tokenizer token, and shared prefixes are represented by shared internal nodes.

model. In this model, the grammar state is derived from LR(1) parser states together with the stack information needed to resolve reductions and goto transitions. Pre[3]-style execution summarizes reduction/goto chains so that cyclic reduction behavior does not lead to non-terminating stack popping for the supported grammars. Therefore, deciding the transition induced by a tokenizer token does not require inspecting an unbounded stack prefix: if $|v|$ is the byte length of the token, $n$ is the number of productions, and $L_{\max}$ is the maximum production right-hand-side length, a conservative bound on the local stack-access depth is

$$d \leq |v| \cdot n \cdot L_{\max} + 1$$

This local boundedness is the property used by GRAM2TOKEN: for a given grammar state, finate stack-top and tokenizer token, the byte-level execution will have a transition outcome, consisting of token validity and the successor runtime grammar state. GRAM2TOKEN does not change the semantic role of runtime grammar states; it pre-computes these token-level outcomes for a given grammar and tokenizer so that decoding can later use table lookups instead of repeatedly resolving byte-level effects at run time.

## 3. GRAM2TOKEN Design

### 3.1. Overview and Design Goals

As discussed in Sections 2.1–2.3, constrained decoding must determine the validity and successor grammar state of to-kenizer tokens whose realizations are byte sequences. Existing systems typically resolve this byte-level grammar effect inside the token-level decoding loop, tightly coupling parser-style grammar execution with auto-regressive token generation at run time.

We propose GRAM2TOKEN, a GPU-native framework that preprocesses the byte-level effect of grammar execution into token-level transition tables before inference. For each pre-processed grammar state, GRAM2TOKEN aligns tokenizer byte sequences with grammar transitions using a tokenizer

trie, producing token-level outcomes that encode both validity and successor-state information. It then groups tokens with identical transition outcomes across grammar states, yielding compact token/category-level tables that reside on the GPU. At run time, each generated token directly indexes these tables, avoiding byte-by-byte grammar traversal on the steady-state critical path. Figure 3 illustrates this separation between preprocessing and GPU-resident decoding.

The remainder of this section presents the design of GRAM2TOKEN. Section 3.2 describes trie-based construction of token-level grammar transitions. Section 3.3 introduces state-aware token categorization for compressing these transitions. Section 3.4 shows how the resulting token/category-level tables are used for GPU-resident constrained decoding.

### 3.2. Token-Level Grammar Representation

Given the runtime grammar state defined in Section 2.3, constrained decoding must determine, for each grammar state $s$ and tokenizer token $v \in \mathcal{V}$, whether emitting $v$ is valid and which successor grammar state it induces. Since token $v$ corresponds to a byte sequence $\phi(v) = (b_1, \ldots, b_k)$, this requires resolving the byte-level grammar effect of the entire sequence $\phi(v)$.

Let $\mathrm{Step}(s, b)$ denote the deterministic one-byte grammar update from runtime grammar state $s$ after consuming byte $b$, including the internal reduction/goto resolution used by the underlying grammar execution model. We write $\mathrm{Exec}(s, w)$ for its extension to a byte string $w$. If the byte string cannot be consumed from $s$, $\mathrm{Exec}(s, w) = \bot$; otherwise it returns the unique successor runtime grammar state. Under the deterministic execution model described in Section 2.3, the transition outcome of a finite token byte sequence is therefore well defined.

GRAM2TOKEN precomputes these token-level outcomes before inference. For each runtime grammar state $s$, we define a token transition table

$$M_s[v] = \mathrm{Exec}(s, \phi(v))$$

where $\bot$ denotes an invalid token. Thus, $M_s[v]$ summarizes the byte-level effect of token $v$ from state $s$. Here, $M_s$ is a semantic view of token-level outcomes; the implementation can stream these outcomes into the categorization procedure during trie traversal without persistently materializing the full state–token matrix.

To construct $M_s$, GRAM2TOKEN traverses the tokenizer trie. For a fixed source state $s$, the trie root is initialized with $s$. Each byte-labeled edge applies one $\mathrm{Step}$ update to the outcome stored at its parent node. Invalid prefixes are assigned $\bot$ and pruned, since no token extending such a prefix can be valid from the same source state. When a

---

**Algorithm 1** Per-State Token-Level Transition Construction

**Input:** source grammar state $s$, byte-level update function $\mathrm{Step}$, tokenizer trie $\mathcal{T}$
**Output:** token transition table $M_s$
Initialize trie root outcome: $O_{\mathrm{root}} \leftarrow s$
**for** each trie depth $d = 1, 2, \ldots$ **do**
  **for all** nodes $u$ at depth $d$ **in parallel do**
    Let $b(u)$ be the byte labeling the edge from parent$(u)$ to $u$
    **if** $O_{\mathrm{parent}(u)} = \bot$ **then**
      $O_u \leftarrow \bot$
    **else**
      $O_u \leftarrow \mathrm{Step}(O_{\mathrm{parent}(u)}, b(u))$
      **if** $O_u$ is invalid **then**
        $O_u \leftarrow \bot$
      **end if**
    **end if**
  **end for**
**end for**
**for all** leaf nodes $u$ corresponding to token $v$ **do**
  $M_s[v] \leftarrow O_u$
**end for**
**return** $M_s$

---

leaf corresponding to token $v$ is reached, the accumulated outcome is recorded as $M_s[v]$.

This trie traversal reuses shared byte prefixes among tokens and can be parallelized across trie nodes at the same depth. It is also independent across source grammar states. The resulting per-state token transition tables are precise but can be large, as they scale with $|\mathcal{S}| \times |\mathcal{V}|$. Section 3.3 introduces state-aware token categorization to compress these tables.

### 3.3. State-Aware Token Categorization

The token-level transition representation in Section 3.2 provides, for each preprocessed grammar state $s \in \mathcal{S}$ and tokenizer token $v \in \mathcal{V}$, a transition outcome $M_s[v]$, which is either an invalid outcome $\bot$ or a successor runtime grammar state. Directly storing and accessing this state–token representation scales as $|\mathcal{S}| \times |\mathcal{V}|$, which is expensive for large vocabularies and grammars with many states.

A natural way to reduce this cost is to group tokens that are indistinguishable from the perspective of grammar execution. Ideally, two tokens $v_i$ and $v_j$ can share one category if they induce identical final transition outcomes from every preprocessed grammar state:

$$v_i \equiv v_j \iff M_s[v_i] = M_s[v_j], \ \forall s \in \mathcal{S}$$

This outcome-equivalence gives the coarsest grammar-preserving categorization: tokens in the same category are either all invalid or lead to the same successor runtime gram-

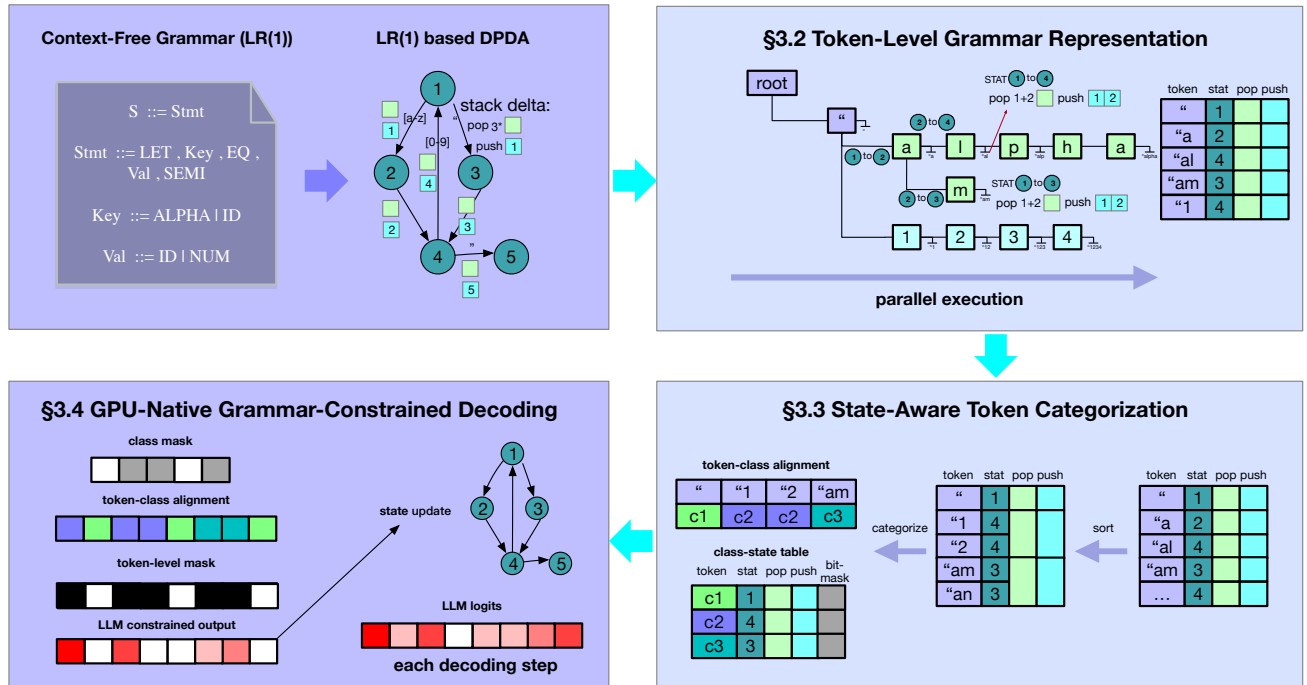

*Figure 3.* Overview of the GRAM2TOKEN framework. Before inference, GRAM2TOKEN aligns byte-level grammar execution with tokenizer byte sequences and compresses the resulting token-level transitions through state-aware token categorization. At run time, generated tokens index GPU-resident token/category-level tables to obtain validity and successor-state information, avoiding byte-by-byte grammar traversal on the steady-state critical path.

mar state for every $s$.

Computing and grouping by the complete final-outcome signature, however, requires explicitly materializing or comparing the full state–token outcome matrix. GRAM2TOKEN instead constructs a conservative refinement of this equivalence during trie-based transition construction. For a source grammar state $s$, the trie-DP traversal computes not only the final outcome $M_s[v]$ at the leaf of token $v$, but also the sequence of intermediate grammar outcomes encountered along the token's byte path. We denote this trace by

$$\tau_s(v) = (o_1, o_2, \ldots, o_{|\phi(v)|})$$

where $o_j$ is the intermediate runtime grammar state, or $\perp$, after consuming the first $j$ bytes of $\phi(v)$ from source state $s$.

GRAM2TOKEN uses these traces to refine token categories incrementally. Initially, all tokens belong to a single category. When processing a grammar state $s$, the trie-DP traversal assigns each token a trace $\tau_s(v)$, and each current category is split according to these traces. Tokens remain in the same category only if their traces are identical for the newly processed state. After all grammar states have been processed, the final categories satisfy

$$v_i \sim v_j \quad \implies \quad \tau_s(v_i) = \tau_s(v_j), \ \forall s \in \mathcal{S}$$

Since identical traces imply identical final outcomes, this trace-based criterion also implies

$$M_s[v_i] = M_s[v_j], \ \forall s \in \mathcal{S}$$

Therefore, the resulting categories are grammar-preserving, although they may be finer than the coarsest outcome-equivalence classes. This conservative refinement avoids unsafe merging while allowing categorization to be fused with trie-DP construction.

Let $\mathcal{C} = \{C_1, \ldots, C_K\}$ denote the final categories, and let $\texttt{cat}(v)$ be the category assigned to token $v$. For each category $C_k$, choose any representative token $r_k \in C_k$. Because all tokens in a category have identical final outcomes for every preprocessed grammar state, we construct a category-level validity table

$$\widetilde{B}[s, k] = \begin{cases} 1 & M_s[r_k] \neq \perp \\ 0 & M_s[r_k] = \perp \end{cases}$$

and a category-level transition table

$$\widetilde{T}[s, k] = M_s[r_k]$$

At run time, token validity and successor-state lookup can therefore be performed at the category level rather than over the full vocabulary.

Algorithm 2 in Appendix D gives the detailed fused trie-DP and partition-refinement procedure. In implementation, GRAM2TOKEN streams trie-DP outcomes into the refinement procedure, maintains outcomes at the block level, and materializes only the final category-level tables. Thus, the full $|\mathcal{S}| \times |\mathcal{V}|$ state–token matrix need not be stored persistently.

## 3.4. GPU-Native Grammar-Constrained Decoding

Building on the token-level transition construction in Section 3.2 and the state-aware categorization in Section 3.3, GRAM2TOKEN performs constrained decoding using GPU-resident token/category-level tables. The run-time state for each request consists of the current runtime grammar state $s_t$ and, in mixed-schema serving, the identifier of the grammar-specific table to use.

At decoding step $t$, each candidate token $v \in \mathcal{V}$ is first mapped to its category $k = \text{cat}(v)$. Given the current grammar state $s_t$, validity is obtained by a table lookup:

$$\text{valid}(v, s_t) = \widetilde{B}[s_t, k]$$

The validity values are expanded to the token vocabulary through the token-to-category map and applied as a mask to the model logits. Equivalently, the constrained distribution follows

$$\tilde{p}(v \mid x_{<t}) \propto \begin{cases} p(v \mid x_{<t}) & \widetilde{B}[s_t, \text{cat}(v)] = 1 \\ 0 & \text{otherwise} \end{cases}$$

After a token $v_t$ is selected, GRAM2TOKEN updates the runtime grammar state by another category-level lookup:

$$s_{t+1} = \widetilde{T}[s_t, \text{cat}(v_t)]$$

Because $\widetilde{T}$ stores the precomputed token-level transition outcome, this update directly applies the byte-level effect of the selected token without re-traversing its byte sequence through the grammar automaton.

All tables used in these operations are resident on the GPU during decoding. Therefore, the steady-state decoding loop uses regular index lookups and elementwise masking rather than parser-style byte traversal or CPU-side grammar execution. This makes grammar handling compatible with batched GPU execution while preserving the same transition semantics encoded in the preprocessed tables.

## 4. Experiments

We evaluate GRAM2TOKEN with a focus on schema-diverse continuous batching, where requests in the same batch may use different structured-output schemas. This setting better reflects multi-tenant structured serving than the shared-grammar setting commonly used in controlled throughput studies. We report the full shared-grammar results in Appendix B.2; the main text focuses on mixed-schema serving, preprocessing amortization, and grammar-complexity scaling.

### 4.1. Experimental Setup

**Tasks and grammars.** We evaluate GRAM2TOKEN on structured generation tasks derived from *JSON-mode-eval* (Nous Research, 2024) and *StructEval* (Cao et al., 2024). JSON-mode-eval evaluates whether model outputs conform to predefined JSON schemas, while StructEval covers grammar-based structured formats including SQL. In the main experiments, we construct batches containing multiple schemas to evaluate schema-diverse continuous batching. Unless otherwise specified, each request enforces its own schema-specific grammar, while all methods use the same tokenizers, and decoding settings.

**Inference framework and models.** All experiments are conducted using the **SGLang** (Zheng et al., 2024) inference framework, which provides GPU-optimized autoregressive decoding. We evaluate four representative open-source LLMs, including LLaMA3-8B (Touvron et al., 2023), Qwen3-8B (Yang et al., 2025), DeepSeek-R1-Distill-Qwen-7B (Guo et al., 2025), and Mistral-7B-Instruct (Jiang et al., 2023). For each model, all decoding methods use the same pretrained checkpoint, tokenizer, batching policy, and hardware setting.

**Baselines and fairness.** We compare GRAM2TOKEN against representative grammar-constrained decoding systems, including XGrammar (Dong et al., 2025), Pre[3] (Chen et al., 2025), and Formatron (Sun et al., 2025). Among them, Pre[3] is the strongest parser-driven baseline in our experiments. To ensure a controlled comparison, the core algorithms of Pre[3] and Formatron are implemented in SGLang using the same tokenizer, batching, and hardware settings as GRAM2TOKEN. For mixed-schema experiments, each method is evaluated under the same schema assignment and request ordering.

**Evaluation metrics.** We report steady-state decoding throughput in tokens per second (TPS) as the primary run-time efficiency metric. Because GRAM2TOKEN moves grammar–token resolution to preprocessing, we also report time-to-first-token (TTFT), preprocessing time, and grammar-table memory where applicable. To characterize end-to-end trade-offs, we further provide break-even analyses that measure when preprocessing overhead is amortized by longer outputs, larger batches, or grammar reuse.

Further implementation details, including precision settings, preprocessing configuration, and memory accounting, are provided in Appendix A.

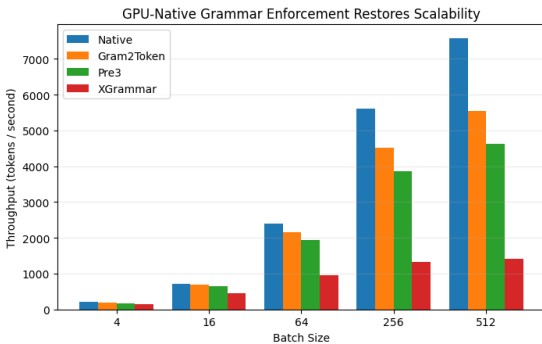

*Figure 4.* Scalability comparison under increasing batch sizes. Gram2Token maintains near-native throughput as batch size grows, while prior grammar-constrained decoding methods exhibit increasing overhead and degraded scalability.

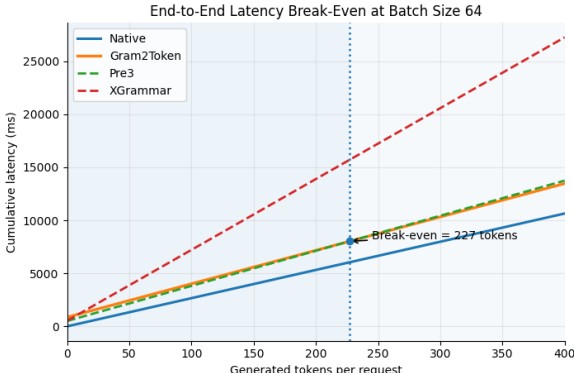

*Figure 5.* End-to-end latency break-even analysis under schema-diverse continuous batching. GRAM2TOKEN incurs higher upfront preprocessing cost, but its lower steady-state per-token overhead amortizes this cost as output length and batch size increase.

### 4.2. Main Results

We first evaluate GRAM2TOKEN in schema-diverse continuous batching, where requests in the same batch are sampled from different JSON schemas. This setting stresses both run-time grammar handling and grammar-table residency, since each request may consult a different schema-specific grammar table. Table 1 reports steady-state throughput, time-to-first-token, and grammar-table memory under representative batch sizes. The full shared-grammar results are provided in Appendix B.2.

Table 1 reports results under schema-diverse continuous batching, where different requests may use different grammar-specific tables. In this harder serving setting, GRAM2TOKEN maintains a steady-state throughput advantage across models and batch sizes. Compared with Pre[3], the strongest parser-driven baseline, GRAM2TOKEN achieves a geometric-mean throughput improvement of $1.38\times$ over the strongest baseline, with a maximum speedup of $1.85\times$, achieves larger gains over CPU-centric baselines. The improvement is more pronounced at larger batch sizes, where parser-style byte traversal and CPU-side grammar handling become harder to scale.

This throughput gain comes with higher TTFT, because GRAM2TOKEN shifts grammar–token resolution from the steady-state decoding loop to preprocessing and GPU-resident lookup tables. Thus, the relevant question is not only whether GRAM2TOKEN improves per-token throughput, but also when the upfront table-construction cost is amortized. Sections 4.3 and Appendix C analyze this trade-off through break-even and grammar-complexity studies.

### 4.3. Preprocessing Amortization

The main results show that GRAM2TOKEN improves steady-state throughput while introducing additional preprocessing and TTFT overhead. We therefore analyze when this upfront

cost is amortized in end-to-end latency. For each batch size, we measure the total latency of generating outputs with increasing lengths and compare GRAM2TOKEN against Pre[3], the strongest parser-driven baseline. We define the break-even length as the minimum number of generated tokens after which GRAM2TOKEN's cumulative latency becomes lower than that of Pre[3].

Figure 5 reports the break-even behavior under schema-diverse continuous batching. Although GRAM2TOKEN starts with higher latency due to table preparation, its lower steady-state per-token cost allows it to catch up as generation proceeds. The break-even length decreases as batch size increases, indicating that larger batches better amortize the preprocessing cost. This result supports the intended use case of GRAM2TOKEN: high-throughput structured serving with recurring grammars and moderately long outputs.

Figure 5 shows that GRAM2TOKEN overtakes Pre[3] after 539, 227, and 46 generated tokens at batch sizes 4, 64, and 512, respectively. This confirms that the preprocessing overhead is most easily amortized in throughput-oriented serving, where requests are batched and outputs are moderately long. In contrast, for cold-start short-output workloads, the additional preparation cost may offset part of the steady-state runtime advantage.

### 4.4. Ablation Study

We ablate the two core components of GRAM2TOKEN on Qwen3-8B at a representative batch size. Token pre-alignment moves byte-level grammar traversal out of the decoding loop by precomputing token-level transition outcomes, while token categorization compresses the resulting transition representation into compact token–category tables. Table 2 reports steady-state throughput and resident transition-table memory.

*Table 1.* Main results under schema-diverse continuous batching. We report steady-state throughput (TPS, higher is better), time-to-first-token (TTFT, ms, lower is better), and resident grammar-table memory (MB, lower is better).

| Model | Method | TPS ↑ | | | TTFT (ms) ↓ | | | Table Mem. (MB) ↓ | | |
|---|---|---|---|---|---|---|---|---|---|---|
| | | B=4 | B=16 | B=64 | B=4 | B=16 | B=64 | B=4 | B=16 | B=64 |
| LLaMA3-8B | XGrammar | 26 | 136 | 581 | 448 | **986** | **2460** | 16 | 193 | 774 |
| | Pre[3] | 30 | 146 | 672 | **408** | 992 | 2898 | 44 | 599 | 3367 |
| | Formatron | 51 | 217 | 772 | 805 | 3604 | 9861 | **8** | **72** | **554** |
| | GRAM2TOKEN | **67** | **298** | **1054** | 545 | 1192 | 5076 | 10 | 94 | 659 |
| Qwen3-8B | XGrammar | 24 | 148 | 579 | 476 | 990 | 2763 | 15 | 187 | 763 |
| | Pre[3] | 28 | 152 | 608 | **398** | **981** | **2298** | 46 | 537 | 3561 |
| | Formatron | 47 | 224 | 771 | 832 | 3757 | 9208 | 9 | **84** | **576** |
| | GRAM2TOKEN | **63** | **271** | **1139** | 473 | 1060 | 4064 | **8** | 92 | 662 |
| DeepSeek-R1-7B | XGrammar | 18 | 127 | 550 | 445 | 967 | 2562 | 14 | 165 | 696 |
| | Pre[3] | 28 | 172 | 641 | **326** | **807** | **2404** | 37 | 601 | 2997 |
| | Formatron | 52 | 248 | 703 | 797 | 4082 | 8760 | 11 | **89** | **604** |
| | GRAM2TOKEN | **66** | **285** | **1303** | 465 | 1352 | 5986 | **8** | 91 | 709 |
| Mistral-7B | XGrammar | 19 | 137 | 558 | 489 | 1022 | 3008 | 13 | 201 | 809 |
| | Pre[3] | 32 | 155 | 612 | **451** | **988** | **2707** | 35 | 508 | 3096 |
| | Formatron | 39 | 196 | 734 | 820 | 2986 | 9547 | **7** | 75 | **492** |
| | GRAM2TOKEN | **44** | **307** | **1206** | 526 | 1322 | 3972 | 9 | 75 | 608 |

*Table 2.* Ablation study on Qwen3-8B at B=64.

| Method | TPS ↑ | TTFT (ms) ↓ | Mem. (MB) ↓ |
|---|---|---|---|
| GRAM2TOKEN full | **1873** | 1060 | **92** |
| w/o token pre-alignment | 1567 | 787 | 862 |
| w/o token categorization | 419 | 865 | 754 |

Table 2 shows that the two components provide complementary benefits. Removing token pre-alignment reduces steady-state throughput, because byte-level grammar effects must be resolved during decoding rather than through precomputed token-level transitions. Removing token categorization substantially increases the size of the resident transition tables and also hurts throughput, indicating that compact token/category-level tables are important not only for memory footprint but also for efficient GPU lookup. Additional mixed-schema memory ablations and grammar-complexity scaling results are provided in Appendices B.1 and C.

### 4.5. Discussion

GRAM2TOKEN changes the performance profile of grammar-constrained decoding by moving grammar–token resolution from the steady-state decoding loop to preprocessing. This introduces additional preparation cost and higher TTFT, but reduces per-token grammar overhead once the token–category tables are resident on the GPU. Our results and break-even analysis show that this trade-off is most favorable in throughput-oriented settings, where grammars are reused, outputs are moderately long, or larger batches are used to improve GPU utilization.

The benefits are less pronounced in latency-critical workloads with very small batches or very short outputs, where the upfront preprocessing cost may offset the steady-state runtime savings. GRAM2TOKEN is therefore best viewed as a design point for reusable structured-output serving, including JSON APIs, SQL generation, and recurring domain-specific formats. The framework assumes that grammar specifications are known before decoding and that the relevant runtime grammar-state space can be preprocessed for the target workload. It does not require changing the semantic role of the underlying deterministic grammar execution model, but it also does not claim to materialize arbitrary unbounded CFG execution offline. For very large, deeply recursive, or dynamically changing grammars, preprocessing cost and table size may increase; practical systems can combine GRAM2TOKEN with parser-driven fallback when execution reaches states outside the preprocessed region.

Our current implementation focuses on standard autoregressive decoding. Since grammar updates are represented as precomputed token/category-level transitions, GRAM2TOKEN is naturally compatible with settings that need to advance or restore grammar states, such as speculative decoding with draft–verify rollback.

# 5. Related Work

## 5.1. Structured Output Generation

Large language models are increasingly used to produce structured outputs such as JSON objects, API function calls, SQL queries, and domain-specific programs (Cai et al., 2023; Park et al., 2024; Li et al., 2024; Cho et al., 2023; Li et al., 2025; Mündler et al., 2025). Unlike free-form text, these outputs are consumed by downstream systems and must satisfy strict syntactic or schema-level constraints. Prior work has explored post-processing, repair, and constrained generation techniques to improve structural validity (Geng et al., 2023; Alshahwan et al., 2024; Park et al., 2025). GRAM2TOKEN focuses on the constrained generation setting, where validity is enforced during decoding rather than repaired after generation.

## 5.2. Grammar-Constrained Decoding for LLMs

Grammar-constrained decoding enforces structural validity by restricting next-token choices according to a predefined formal grammar (Geng et al., 2023). Existing systems typically compile grammars into executable automata and track grammar states during decoding to construct validity masks. XGrammar (Dong et al., 2025) represents one line of automata-based constrained decoding, while Pre[3] (Chen et al., 2025) improves runtime efficiency through deterministic grammar execution and precomputed parser information. Formatron (Sun et al., 2025) further explores pruning-based parsing mechanisms to reduce runtime search. These methods improve the efficiency and reliability of constrained decoding, but grammar execution remains coupled to the token-level decoding loop: each decoding step still needs to resolve token validity and successor grammar states through parser-style traversal or runtime grammar logic.

GRAM2TOKEN is complementary to these parser-driven systems. Rather than optimizing byte-level grammar execution inside the decoding loop, it preprocesses the byte-level effect of tokenizer tokens into token/category-level transition tables before inference. At run time, grammar handling is reduced to GPU-resident lookup, masking, and state update over preprocessed tables. This shifts the main cost from steady-state decoding to preprocessing, making the approach especially suitable for reusable grammars and throughput-oriented structured serving.

## 5.3. GPU-Efficient Decoding and System-Level Acceleration

A complementary line of work optimizes LLM inference on modern accelerators, including fused kernels (Li et al., 2022), memory-efficient attention mechanisms (Dao et al., 2022; Dao, 2023), and high-throughput serving frameworks such as SGLang (Zheng et al., 2024), vLLM (Kwon, 2025), and FasterTransformer (Chelba et al., 2020). These systems improve GPU utilization through batching, scheduling, memory management, and kernel-level optimization. However, they are largely orthogonal to grammar-constrained decoding: generic inference optimizations do not remove the semantic mismatch between byte-level grammar execution and tokenizer-level generation. GRAM2TOKEN addresses this mismatch directly by converting grammar handling into GPU-resident token/category-level table operations, allowing constrained decoding to better align with batched GPU execution.

# 6. Conclusion

This paper introduced GRAM2TOKEN, a GPU-native framework for high-throughput grammar-constrained decoding. GRAM2TOKEN preprocesses deterministic byte-level grammar execution into token-level transition summaries, aligns tokenizer byte sequences with grammar transitions through a trie, and compresses the resulting representation with state-aware token categorization. At run time, each generated token indexes GPU-resident token/category-level tables for validity and successor-state lookup, avoiding parser-style byte traversal on the steady-state critical path.

Experiments on structured JSON and SQL generation show that GRAM2TOKEN achieves a geometric-mean throughput improvement of $1.38\times$ over the strongest baseline, with a maximum speedup of $1.85\times$ and achieves larger gains over CPU-centric baselines, while introducing additional preprocessing and time-to-first-token overhead. Break-even, mixed-schema, and grammar-complexity analyses further characterize when this cost is amortized. These results position token-level grammar preprocessing as an effective design point for throughput-oriented structured LLM serving with reusable grammars, moderately long outputs, and large batches.

## Acknowledgements

We thank the anonymous reviewers and area chairs for their constructive feedback. This work was supported by the National Natural Science Foundation of China (NSFC) under Grant No. 62503491.

## Impact Statement

This paper presents a systems-oriented contribution whose primary goal is to improve the efficiency and scalability of grammar-constrained decoding for large language models. By reducing the runtime overhead of enforcing formal grammars during generation, GRAM2TOKEN aims to make structured outputs more reliable and efficient in practical deployment settings.

The proposed techniques do not train, modify, or add new capabilities to the underlying language model. Instead, they optimize the execution of existing grammar constraints during decoding. As such, we do not anticipate novel ethical risks beyond those already associated with the deployment of large language models, including misuse, bias, privacy leakage, and unsafe automation. More efficient constrained decoding could lower the cost of large-scale structured generation or automated tool/API interaction, so deployments should still apply appropriate access control, monitoring, and application-specific safety checks.

We also expect potential positive impacts. Faster and more reliable constrained decoding can make language-model outputs more predictable in applications that require strict formats, such as API calls, program synthesis, SQL generation, and structured data extraction. Overall, the societal impact of this work is aligned with improving the efficiency, reliability, and controllability of machine learning systems.

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

# A. Experimental Details and Reproducibility

This appendix provides implementation and measurement details that are omitted from the main text due to space constraints.

## A.1. System Configuration

We implement GRAM2TOKEN as a GPU-resident constrained decoding module within the SGLang inference framework (Zheng et al., 2024). The module interfaces with SGLang's decoding loop through its `CustomLogitsProcessor` abstraction, allowing grammar masks and grammar-state updates to be applied during decoding without invoking a CPU-side parser on the steady-state critical path.

All experiments are conducted on a server with dual AMD EPYC 9554 CPUs and four NVIDIA A800 GPUs, each with 80GB of GPU memory, interconnected via PCIe. Unless otherwise specified, all methods use the same model checkpoint, tokenizer, hardware configuration, and SGLang scheduling policy.

## A.2. Decoding and Measurement Protocol

All experiments use greedy decoding with temperature set to $0$ and top-$p$ set to $1.0$. This removes sampling variance and ensures that performance measurements reflect the efficiency of grammar-constrained decoding rather than stochastic sampling behavior. All methods are evaluated under SGLang's continuous batching and RadixAttention-based prefix caching. Requests in a batch may have different sequence lengths and runtime grammar states; in schema-diverse experiments, requests may additionally use different grammar-specific tables.

Decoding throughput is reported in tokens per second (TPS). For a batch of size $B$, TPS is computed as

$$\text{TPS} = \frac{\sum_{i=1}^{B} T_i}{\Delta t}$$

where $T_i$ is the number of generated tokens for request $i$, and $\Delta t$ is the wall-clock decoding time measured from the start of token generation to the completion of the batch. Time-to-first-token (TTFT) is measured as the elapsed time from request submission to the emission of the first generated token. For GRAM2TOKEN, TTFT includes grammar preprocessing and table preparation when the corresponding grammar tables are not already resident.

## A.3. Grammar Preprocessing and Table Residency

Grammar preprocessing in GRAM2TOKEN includes deterministic grammar execution setup, trie-based token transition construction, and state-aware token categorization. The resulting GPU-resident data structures include token-to-category mappings, category-level validity tables, and category-level transition tables. These tables are prepared per grammar and tokenizer configuration and can be reused across requests that share the same grammar.

For shared-grammar experiments, a single grammar table is resident and reused by all requests in a batch. For schema-diverse experiments, multiple grammar-specific tables may be resident simultaneously, and each request carries the identifier of the table used for lookup. Reported table memory corresponds to the resident grammar tables used by the evaluated setting, excluding model weights and KV cache memory.

## A.4. Baseline Implementations

We compare against XGrammar (Dong et al., 2025), Pre[3] (Chen et al., 2025), and Formatron (Sun et al., 2025). Pre[3] and Formatron are reimplemented within the SGLang environment so that they share the same tokenizer, batching strategy, KV-cache management, and hardware configuration as GRAM2TOKEN. XGrammar is evaluated using its official integration. All methods enforce the same grammars on the same prompts without post-hoc filtering or correction. We tune batching-related parameters for each method to use its peak stable throughput under the evaluated setting.

# B. Additional Experimental Results

This appendix provides additional experimental results that complement the main text. B.1 reports detailed schema-diverse continuous-batching results, including throughput, TTFT, and resident grammar-table memory. B.2 reports representative shared-grammar results. B.3 isolates the cost of grammar enforcement from LLM forward computation.

*Table 3.* Detailed schema-diverse continuous-batching results on JSON-mode-eval. TPS is higher better; TTFT and memory are lower better.

| Batch | TPS ↑ | | TTFT (ms) ↓ | | Resident Mem. (MB) ↓ | | Mem. Red. |
|---|---|---|---|---|---|---|---|
| | GRAM2TOKEN | Pre[3] | GRAM2TOKEN | Pre[3] | Full | w/o Categ. | |
| 4 | 63 | 28 | 473 | 398 | 8 | 36 | 4.5× |
| 16 | 271 | 152 | 1060 | 981 | 92 | 754 | 8.2× |
| 64 | 1139 | 608 | 4064 | 2298 | 662 | 1270 | 1.9× |

*Table 4.* Representative shared-grammar decoding efficiency. We report throughput (TPS, higher is better) and time-to-first-token (TTFT, ms, lower is better).

| Model | Method | TPS ↑ | | | TTFT (ms) ↓ | | |
|---|---|---|---|---|---|---|---|
| | | B=4 | B=64 | B=512 | B=8 | B=64 | B=512 |
| LLaMA3-8B | XGrammar | 144 | 958 | 1424 | 484 | 534 | 592 |
| | Pre[3] | 176 | 1934 | 4625 | 453 | 506 | 560 |
| | Formatron | 126 | 742 | 1037 | 515 | 568 | 621 |
| | GRAM2TOKEN | **187** | **2150** | **5536** | 810 | 863 | 921 |
| Qwen3-8B | XGrammar | 124 | 821 | 1221 | 525 | 583 | 644 |
| | Pre[3] | 148 | 1669 | 4155 | 494 | 551 | 612 |
| | Formatron | 109 | 637 | 886 | 556 | 612 | 672 |
| | GRAM2TOKEN | **156** | **1836** | **4930** | 854 | 911 | 972 |
| DeepSeek-R1-7B | XGrammar | 171 | 1153 | 1733 | 440 | 490 | 539 |
| | Pre[3] | 212 | 2269 | 5212 | 410 | 458 | 507 |
| | Formatron | 147 | 894 | 1269 | 469 | 520 | 565 |
| | GRAM2TOKEN | **228** | **2553** | **6281** | 774 | 822 | 867 |
| Mistral-7B | XGrammar | 232 | 2040 | 3733 | 377 | 420 | 457 |
| | Pre[3] | 291 | 3053 | 6833 | 351 | 388 | 429 |
| | Formatron | 195 | 1660 | 2948 | 411 | 452 | 487 |
| | GRAM2TOKEN | **320** | **3409** | **7779** | 710 | 752 | 793 |

## B.1. Detailed Schema-Diverse Results

Table 3 reports detailed results under schema-diverse continuous batching on JSON-mode-eval. Requests in this setting are sampled from different schema-specific grammars, so each request may consult a different resident grammar table. We report both runtime performance and resident grammar-table memory. The memory ablation compares the full GRAM2TOKEN system against a variant without token categorization.

These results show that GRAM2TOKEN retains a steady-state throughput advantage even when requests use different schema-specific grammars. The resident memory results also show that token categorization remains important in schema-diverse batching: although total table memory grows with the number of resident grammars, categorization substantially reduces the footprint of the resident grammar tables.

## B.2. Representative Shared-Grammar Results

Table 4 reports representative shared-grammar continuous-batching results. In this setting, requests in a batch share the same grammar table, while sequence lengths and runtime grammar states may differ across requests. These results complement the schema-diverse experiments by showing the controlled setting used in the original batch-scaling evaluation.

*Table 5.* Grammar-only throughput across batch sizes, excluding LLM forward computation. Throughput is measured in token-level grammar updates per second.

| Batch | Pre[3] TPS | XGrammar TPS | Formatron TPS | GRAM2TOKEN TPS ↑ |
|---|---|---|---|---|
| 4 | 2,941 | 330 | 3,198 | **42,018** |
| 16 | 6,556 | 479 | 8,282 | **75,158** |
| 64 | 9,465 | 540 | 10,316 | **142,290** |
| 256 | 10,646 | 558 | 11,325 | **167,473** |
| 512 | 10,872 | 561 | 12,703 | **182,406** |

*Table 6.* Grammar-only processing statistics at batch size 512. Memory reports resident grammar-related table memory, excluding model weights and KV cache.

| Method | Pure Logic Throughput (TPS) ↑ | Extra Memory (MB) ↓ | TTFT Prep. (ms) ↓ |
|---|---|---|---|
| XGrammar | 561 | 35.58 | 643 |
| Pre[3] | 10,872 | 85 | 615 |
| Formatron | 12,703 | 120 | 667 |
| GRAM2TOKEN | **182,406** | **1.36** | 931 |

## B.3. Grammar-Only Throughput and Memory

We further isolate the cost of grammar enforcement from LLM forward computation. This microbenchmark measures how many token-level grammar validity and state-update operations each method can process per second, excluding model forward time. The goal is to compare the raw efficiency of the grammar engines themselves, independent of differences in model execution, attention computation, or KV-cache management.

Table 5 shows a large gap between GRAM2TOKEN and parser-driven baselines when LLM computation is removed. This confirms that the main runtime advantage of GRAM2TOKEN comes from changing the structure of grammar enforcement: instead of resolving byte-level grammar effects through parser-style traversal at each decoding step, the runtime path uses preprocessed token/category-level tables for validity lookup and state update. The advantage becomes more visible as batch size increases, where regular GPU-resident lookup and masking better match batched execution.

Table 6 reports the corresponding memory and preprocessing statistics at batch size 512. GRAM2TOKEN has higher preprocessing-related TTFT than the parser-driven baselines, reflecting the cost of trie-based token transition construction and state-aware token categorization. However, the resulting resident table footprint is small in the shared-grammar setting because token categorization compresses many vocabulary tokens into behavior-equivalent categories. This reduction is important for keeping grammar tables resident on the GPU and for making the runtime lookup path efficient.

Overall, these grammar-only measurements support the system design of GRAM2TOKEN. The method trades additional upfront preprocessing for substantially lower steady-state grammar-processing cost. This trade-off is most favorable when grammar tables are reused across requests, when outputs are long enough to amortize preprocessing, or when large batches make per-token runtime overhead the dominant bottleneck.

# C. Grammar-Complexity Scaling

This appendix studies how GRAM2TOKEN's preprocessing cost and resident table memory scale with grammar complexity. Since GRAM2TOKEN moves grammar–token resolution from the steady-state decoding loop to preprocessing, more complex grammars may increase the number of preprocessed grammar states, preprocessing time, and resident table memory. We therefore report controlled measurements over JSON schemas with increasing nesting depth and a larger programming-language grammar stress test.

## C.1. JSON Schema Complexity

We first group JSON-mode-eval schemas by maximum nesting depth. Table 7 reports the number of schemas in each group, average depth, preprocessing time, resident table memory, and number of token categories.

Table 7 shows that preprocessing time and the number of token categories increase with schema depth, while resident table memory grows only mildly in this benchmark. This indicates that deeper JSON schemas introduce more distinct grammar behavior, but token categorization keeps the resident table representation compact.

*Table 7.* JSON-mode-eval complexity scaling by schema depth. Memory is resident grammar-table memory in MB.

| Depth Group | $n$ | Avg. Depth | Prep. (ms) | Mem. (MB) | Categories |
|---|---|---|---|---|---|
| Depth = 0 | 6 | 0.00 | 339.67 | 0.8881 | 3.00 |
| Depth = 1–2 | 84 | 1.42 | 377.53 | 0.8966 | 20.67 |
| Depth = 3–4 | 10 | 3.10 | 423.29 | 0.9061 | 29.70 |

*Table 8.* JSON-mode-eval complexity scaling by reachable grammar states. Memory is resident grammar-table memory in MB.

| Reachable States | $n$ | Prep. (ms) | Mem. (MB) | Categories |
|---|---|---|---|---|
| $[0, 16)$ | 8 | 334.86 | 0.8882 | 4.00 |
| $[16, 32)$ | 16 | 351.30 | 0.8920 | 14.56 |
| $[32, 64)$ | 60 | 381.00 | 0.8967 | 21.85 |
| $[64, 128)$ | 16 | 426.46 | 0.9075 | 29.69 |

We also group the same schemas by the number of reachable grammar states, which more directly reflects the preprocessed state space. Table 8 shows the corresponding scaling behavior.

The reachable-state grouping shows the same trend: larger preprocessed state spaces increase preprocessing time and token-category diversity. The memory increase remains modest because the category-level representation avoids storing a full state–token table.

### C.2. Programming-Language Grammar Stress Test

We additionally evaluate a larger grammar derived from the C language grammar specification. This experiment is intended as a stress test for a substantially more complex grammar than the JSON schemas used in the main benchmark. It characterizes preprocessing and resident-memory cost under a larger deterministic grammar, rather than claiming that arbitrary unbounded programming-language grammars can always be fully materialized without limits.

Table 9 shows that moving from a smaller C subset to a larger raw Annex A.2 grammar substantially increases the number of reachable states, TTFT, and resident table memory. At the same time, once the relevant tables are prepared, the steady-state throughput remains comparable. This supports the design trade-off of GRAM2TOKEN: grammar complexity mainly affects preprocessing and table residency, while the steady-state decoding loop continues to use the same GPU-resident lookup path.

### C.3. Scope of Preprocessing

These results clarify the intended scope of GRAM2TOKEN. The method is designed for deterministic, reusable structured grammars whose relevant runtime grammar-state space can be preprocessed for the target workload. It does not claim to materialize arbitrary unbounded CFG execution offline. For very large, deeply recursive, or dynamically changing grammars, practical deployments may combine GRAM2TOKEN with parser-driven fallback when execution reaches states outside the preprocessed region.

## D. Fused Token Categorization Algorithm

This appendix gives the detailed fused trie-DP and partition-refinement procedure used for state-aware token categorization. As discussed in Section 3.3, the implementation constructs a conservative refinement of final-outcome equivalence. During trie traversal, each token obtains a trace of intermediate grammar outcomes from a source grammar state. Categories are refined according to these traces, so tokens with different byte-level execution traces are placed into different categories even when their final successor states coincide. This may produce more categories than the coarsest outcome-equivalence partition, but it preserves grammar semantics because identical traces imply identical final outcomes.

*Table 9.* C grammar stress test at batch size 64. Memory is resident grammar-table memory.

| Grammar | Depth | #Prod. | #States | TTFT (ms) | Mem. (MB) | TPS ↑ |
|---|---|---|---|---|---|---|
| Full raw Annex A.2 | 12 | 370 | 4116 | 1758.39 | 1.0804 | 1998 |
| Minimal subset | 11 | 93 | 173 | 1254.12 | 0.1650 | 1902 |

---

**Algorithm 2** Fused Trie-DP Token Categorization

---

**Input:** grammar states $\mathcal{S} = \{s_1, \ldots, s_m\}$, tokenizer trie $\mathcal{T}$, byte-level update function $\text{Step}$
**Output:** token category map $\texttt{cat}$, validity table $\widetilde{B}$, transition table $\widetilde{T}$
Initialize partition $\mathcal{P} \leftarrow \{\mathcal{V}\}$
**for** $i = 1$ to $m$ **do**
    Initialize root outcome $O_{\text{root}} \leftarrow s_i$
    Initialize root trace key $K_{\text{root}} \leftarrow ()$
    **for** each trie depth $d = 1, 2, \ldots$ **do**
        **for all** nodes $u$ at depth $d$ **in parallel do**
            Let $b(u)$ be the byte labeling the edge from $\text{parent}(u)$ to $u$
            **if** $O_{\text{parent}(u)} = \bot$ **then**
                $O_u \leftarrow \bot$
            **else**
                $O_u \leftarrow \text{Step}(O_{\text{parent}(u)}, b(u))$
                **if** $O_u$ is invalid **then**
                    $O_u \leftarrow \bot$
                **end if**
            **end if**
            $K_u \leftarrow \text{Append}(K_{\text{parent}(u)}, O_u)$
        **end for**
    **end for**
    Initialize empty refinement map $\mathcal{H}$
    **for all** leaf nodes $u$ corresponding to token $v$ **do**
        Let $C(v)$ be the current block containing $v$
        Insert $v$ into $\mathcal{H}[(C(v), K_u)]$
        Record final outcome $O_u$ for token $v$ under state $s_i$
    **end for**
    Refine $\mathcal{P}$ using the nonempty token sets in $\mathcal{H}$
**end for**
**for all** final categories $C_k \in \mathcal{P}$ **do**
    Choose a representative token $r_k \in C_k$
    **for all** tokens $v \in C_k$ **do**
        $\texttt{cat}(v) \leftarrow k$
    **end for**
    **for** $i = 1$ to $m$ **do**
        Let $o_i$ be the recorded final outcome of $r_k$ under state $s_i$
        **if** $o_i = \bot$ **then**
            $\widetilde{B}[s_i, k] \leftarrow 0$
            $\widetilde{T}[s_i, k] \leftarrow \bot$
        **else**
            $\widetilde{B}[s_i, k] \leftarrow 1$
            $\widetilde{T}[s_i, k] \leftarrow o_i$
        **end if**
    **end for**
**end for**
**return** $\texttt{cat}, \widetilde{B}, \widetilde{T}$

---

By induction over the processed grammar states, after processing $s_1, \ldots, s_i$, each block in $\mathcal{P}$ contains only tokens whose trace signatures are identical for these $i$ states. After all states are processed, tokens in the same final category therefore have identical traces for every preprocessed grammar state. Since identical traces imply identical final transition outcomes, the final categories are safe for category-level validity and successor-state lookup.

## E. Preprocessing Cost Breakdown

GRAM2TOKEN moves grammar–token resolution from the steady-state decoding loop to preprocessing. This preprocessing consists of three steps: deterministic grammar execution setup, trie-based token transition construction, and state-aware token categorization. The resulting token-to-category map, category-level validity table, and category-level transition table are then materialized in GPU-resident memory.

The additional TTFT reported for GRAM2TOKEN includes this preprocessing and table preparation when grammar tables are not already resident. This cost is paid per grammar–tokenizer configuration and can be reused across requests that share the same grammar. Once preprocessing completes, decoding does not perform parser-style byte traversal on the steady-state critical path; it performs category lookup, validity masking, and successor-state update using the preprocessed tables.

This design is therefore most useful when the preprocessing cost can be amortized through grammar reuse, moderately long outputs, or larger batches. For workloads with frequently changing grammars and very short outputs, parser-driven methods may have lower cold-start latency.

## F. Correctness and Scope

We clarify the correctness guarantee of GRAM2TOKEN. The guarantee is defined over the finite set of runtime grammar states $\mathcal{S}$ that are preprocessed for a given grammar and tokenizer. For each $s \in \mathcal{S}$ and tokenizer token $v \in \mathcal{V}$, GRAM2TOKEN records the token-level outcome $M_s[v]$, which is either $\perp$ for an invalid token or the successor runtime grammar state obtained by executing the underlying deterministic byte-level grammar update over $\phi(v)$.

Let $\mathrm{Exec}(s, w)$ denote the deterministic byte-level execution from state $s$ on byte string $w$. During preprocessing, GRAM2TOKEN constructs $M_s[v]$ so that
$$M_s[v] = \mathrm{Exec}(s, \phi(v))$$

Therefore, for all preprocessed states and tokenizer tokens, token validity and successor-state updates match the underlying byte-level execution.

State-aware token categorization preserves this property. In the implementation, categories are obtained by conservative trace-based refinement: tokens are grouped only when their trie-DP execution traces match for all processed states. Identical traces imply identical final token-level outcomes, so replacing tokens by categories does not merge tokens with different validity or successor-state behavior over $\mathcal{S}$.

The guarantee is therefore exact over the preprocessed state space $\mathcal{S}$. It should not be interpreted as a claim that arbitrary unbounded CFG execution can always be fully materialized offline. For grammars or outputs whose execution reaches states outside the preprocessed region, a practical deployment can fall back to parser-driven decoding or extend preprocessing bounds. This matches the intended use of GRAM2TOKEN: reusable structured grammars whose relevant runtime state space can be preprocessed for the target workload.

