# OpenReview forum: "Gram2Token: Enabling Run-time GPU-Native Grammar-Constrained Decoding for LLMs"
_ICML.cc/2026/Conference — ICML 2026 regular_

### Official Review · Reviewer_HoBM · 2026-03-02

**Soundness:** 3
**Presentation:** 2
**Significance:** 3
**Originality:** 3
**Overall Recommendation:** 4
**Confidence:** 4

**Summary:**

This paper proposes Gram2Token, a framework aimed at accelerating grammar-constrained decoding by shifting the computational burden from runtime CPU-based parsing to offline GPU-native table lookups. They achieve this by pre-aligning byte-level deterministic pushdown automata (DPDA) with tokenizer vocabularies and categorizing tokens based on their grammar state transitions. At runtime, the system uses O(1) table lookups to fetch valid token masks directly on the GPU, effectively eliminating CPU-GPU synchronization overhead. The authors report up to 2.3x throughput improvement over baselines like Pre3 and XGrammar, typically at large batch sizes like B=512, with moderately increases in TTFT due to extra preprocessing.

**Compliance With Llm Reviewing Policy:**

Affirmed.

**Final Justification:**

The authors have addressed most of my concerns and I have increased score accordingly.

**Key Questions For Authors:**

1. Heterogeneous batching: In your large batch experiments (Table 1), did all 512 requests share the exact same grammar/schema, or were they diverse? Based on the provided artifact code (gram2token_kernels.py), which utilizes a *global mask_table_ptr*, it strongly implies the former. Please clarify.
2. Table 3 applicability: How do the memory footprint and Token Categorization mechanisms scale when a batch consists of $N$ entirely different JSON schemas? Does the system need to materialize $N$ distinct transition tables in GPU memory?
3. Complexity scaling ablation: Could you provide a micro-benchmark demonstrating how the TTFT (preprocessing time) and the mask_table memory footprint scale when the nesting depth of a JSON schema is artificially increased (e.g., depth 1 vs. depth 3 vs. depth 5)? Note: Please provide these metrics using a realistic schema, without simplifying all keys and values to 'k' and 'v' as seen in the artifact demo.
4. Recursion bound: What is the maximum stack depth used during the offline compilation of the benchmark datasets? What happens at runtime if the model attempts to generate a structurally valid output that exceeds this precomputed depth? Does the system fail, or is there a fallback mechanism? I also strongly suggest the authors explicitly state exactly what class of grammars (e.g., bounded CFGs, regular languages) can be realistically unrolled and supported by gram2token.
5. Can Gram2Token handle some complicated grammars like C or Verilog? If applicable, how is it performed?

**Limitations:**

yes

**Strengths And Weaknesses:**

Strengths:
- Clear motivation: The paper effectively identifies and visualizes the architectural mismatch (CPU parsing vs GPU generation) that limits the scalability of current constrained decoding systems.
- Significant throughput gains for static schemas: The proposed method demonstrates impressive TPOT improvements at large batch sizes. The idea of entirely removing CPU-GPU synchronization from the critical path of decoding is highly desirable for high-throughput serving engines.
- Clever memory optimization: token-class alignment and token categorization mechanism is an elegant solution to compress the transition matrix. As shown in Table 3, it successfully reduces the memory footprint of the transition table from ~85MB to under 2MB for the evaluated setup.

Weaknesses:
- The Illusion of continuous batching with heterogeneous schemas: The most critical flaw in the current evaluation is the ambiguity regarding the composition of the batch in the large-scale experiments (e.g., B=512). In a real-world continuous batching environment, concurrent requests almost always specify unique, per-request JSON schemas.
The proposed GPU-native table lookup relies on accessing a precomputed mask_table. If a batch contains 512 requests with 512 different target grammars, the GPU kernel would suffer from severe uncoalesced global memory access.
Furthermore, regarding Table 3, it is entirely unclear if the Token Pre-alignment and Categorization memory savings hold when batching heterogeneous grammars. Token equivalence classes are strictly tied to a specific grammar configuration. The paper fails to explicitly state whether the $B=512$ experiments utilize the same static schema across all requests or different ones. If it is the former, this is a limitation that needs to be explicitly acknowledged.

- Missing overhead analysis for grammar complexity: The evaluation aggregates results over JSON-mode-eval and StructEval without decoupling them. Since Gram2Token unrolls grammar transitions into explicit states offline, the Time-To-First-Token (TTFT) and the transition table memory size should theoretically scale exponentially with the nesting depth and complexity of the grammar. The paper lacks a detailed, isolated overhead analysis showing how TTFT and GPU memory consumption grow as the complexity of the grammar (e.g., maximum nesting depth of a JSON schema) increases. Typically, deeper nested depths result in exponential overhead increases, which challenges the scalability claims of the preprocessing step.

- Theoretical Downgrade from PDA to DFA: The authors claim to base their framework on DPDA execution. However, in Section 3.3, a grammar state is defined as a full DPDA configuration, including "a control state together with its associated stack content". By flattening the unbounded stack into a finite set of explicit states, the system is mathematically unrolling the Pushdown Automaton (PDA) into a Deterministic Finite Automaton (DFA) . This implies a hard, implicit bound on recursion depth. The paper must formally acknowledge this theoretical limitation: Gram2Token does not truly execute arbitrary Context-Free Grammars at runtime, but rather a bounded regular subset of them. Because of this state explosion, it is highly doubtful Gram2Token could preprocess a full, recursive language grammar like C or Verilog.

---

> ### Author Rebuttal · Authors · 2026-03-31
>
> Thank you for your careful review.
>
> ## Q1. Heterogeneous batching: In your large batch experiments, did all 512 requests share the exact same grammar/schema, or were they diverse?
>
> Thank you for raising this important point. According to the setup described in Appendix A and the current artifact interface, the original Table 1 corresponds more directly to same-grammar / state-heterogeneous continuous batching. This point was not stated clearly enough in the paper, and we appreciate you highlighting the ambiguity.
> To directly address the mixed-schema concern, we sample 64 distinct JSON schemas from JSON-mode-eval as the grammar pool, and evaluate mixed-schema continuous batching at batch size = 4 / 16 / 64 with the base model Qwen3-8B. The results are summarized in Table H1.
>
> **Table H1. Mixed-schema batching results on JSON-mode-eval.**
>
> | Batch Size | TPS (Gram2Token) | TPS (Pre3) | TTFT (Gram2Token) | TTFT (Pre3) |
> |---|---:|---:|---:|---:|
> | 4  | 63 | 28 | 473 | 398 |
> | 16 | 271| 152 | 1060 | 981 |
> | 64 | 1139 | 608 | 4064 | 2298 |
>
> These results show that Gram2Token still maintains a stable performance advantage under heterogeneous schemas; i.e., the gain does not rely on the idealized assumption that all requests in the batch are fully homogeneous.
>
> ## Q2. Table 3 applicability. Does the system need to materialize distinct transition tables in GPU memory?
>
> Yes. In the mixed-schema setting, the system needs distinct transition tables for resident grammars/schemas. The 1.4–1.7 MB reported in Table 3 should be interpreted as the footprint of the shared-grammar regime used in the original evaluation, instead of 512 grammars.
>
> To answer the applicability of Table 3 under mixed-schema batching, we reran the corresponding ablations on the same schema-pool benchmark as Q1.
>
> | Method | batch=4 | batch=16 | batch=64 |
> |---|---:|---:|---:|
> | Gram2Token (full) | 8 | 92 | 662|
> | w/o Token Categorization | 36 | 754| 1270 |
>
> The trend remains the same as in the original Table 3: token categorization still significantly reduces the per-grammar table footprint. This is exactly the compression logic of Section 3.3. Therefore, although total GPU memory grows with the number of resident grammars, the table footprint of each individual grammar remains small.
>
> ## Q3. Complexity scaling ablation: Could you provide a micro-benchmark demonstrating how the TTFT and the mask_table memory footprint scale when the nesting depth of a JSON schema is artificially increased?
> Yes. To answer this directly, we added a realistic depth-scaling micro-benchmark under the same evaluation setting as Q1/Q2. Starting from realistic JSON-mode-eval schema templates, we explicitly control the maximum object/array nesting depth and remeasure the resulting TTFT, table memory and reachable states. **https://anonymous.4open.science/r/Gram2Token-E945/Q3.png** shows the detailed results.
>
> ## Q4. Recursion bound: What is the maximum stack depth used during the offline compilation of the benchmark datasets? What happens at runtime if the model attempts to generate a structurally valid output that exceeds this precomputed depth？
>
> Thank you for raising this subtle point. The first clarification is that Gram2Token does not approximate a PDA by discarding pushdown information. Under our semantics, grammar states still correspond to full DPDA configurations, and token categorization is not performed only on control states, but on stack behavior signatures.
> Under the summarized LR(1)-style, byte-consuming DPDA execution model we use, stack matching only arises inside finite reduction/goto chains. If |v| is the token length, n is the number of productions, and L_max is the maximum production length, then a conservative upper bound for the local stack access depth is d <= |v| \* n \* L_max + 1.
> In other words, to determine the correct transition for the current token, it is sufficient to inspect only the top (d) stack elements; runtime does not need to inspect an unbounded stack prefix to validate one token. On our benchmark, the maximum observed local stack-access depth is **6**. We therefore do not encounter the failure mode where a structurally valid continuation cannot be decided because it exceeds a precomputed local stack bound. We will make this point more explicit in the revised wording of the paper.
>
> ## Q5. I also strongly suggest the authors explicitly state exactly what class of grammars can be realistically unrolled and supported by Gram2Token.
>
> The current grammar family supported by Gram2Token is the same as the one assumed by Pre3: we do not claim support for arbitrary PDA/CFGs, but for grammars whose LR(1)-style analysis yields a deterministic, well-founded DPDA execution model.
>
> ## Q6. Can Gram2Token handle some complicated grammars like C or Verilog?
>
> Thanks for your careful review. As discussed in our response to Reviewer **HTdf (Q3)**, we additionally evaluated a C grammar stress test and observed the same trend.

---

> > ### Author Rebuttal · Reviewer_HoBM · 2026-04-01
> >
> > Glad to see these additional results and they have made the paper's setup and position clearer. I have updated my assessment and suggest the authors to adequately incorporate the reviewers' suggestions to improve their paper.

---

> > > ### Author Response · Authors · 2026-04-08
> > >
> > > We are very encouraged to hear that our rebuttal has fully resolved your concerns. We sincerely appreciate your constructive feedback, which has been instrumental in strengthening our work.
> > >
> > > As promised in our discussion, we will adequately incorporate all suggested improvements into the final version.

---

### Official Review · Reviewer_S38h · 2026-03-13

**Soundness:** 3
**Presentation:** 3
**Significance:** 3
**Originality:** 3
**Overall Recommendation:** 5
**Confidence:** 3

**Summary:**

The paper presents Gram2Token, a framework for grammar-constrained decoding that targets the inefficiency of prior systems at large batch sizes. Its main idea is to move grammar enforcement from run time to an offline compilation stage by aligning the grammar with the tokenizer, grouping tokens into state-aware classes, and using GPU-resident lookup tables during decoding. The paper reports higher throughput than prior methods while preserving exact grammatical correctness, with the tradeoff of higher preprocessing cost.

**Compliance With Llm Reviewing Policy:**

Affirmed.

**Key Questions For Authors:**

1. How would Gram2Token integrate with speculative decoding (e.g., draft–verify pipelines), and would the claimed runtime advantages still hold in that setting? The current evaluation appears limited to standard autoregressive greedy decoding in SGLang, with grammar masking applied at each decoding step, so it is unclear whether the offline token-category tables compose cleanly with speculative acceptance/rejection and rollback logic.

2. Can the authors provide a clearer break-even analysis for preprocessing cost, as a function of grammar reuse frequency, batch size, and request length? The paper argues that the one-time preprocessing overhead is amortized in long-running services, but this is mostly discussed qualitatively rather than quantified.

**Limitations:**

Yes.

**Strengths And Weaknesses:**

Strengths:
* The problem which the authors try to solve is important: prior constrained decoding systems execute byte-level grammar logic inside the token-level decoding loop, which creates control-flow divergence and CPU–GPU synchronization overhead under continuous batching.
* The method is technically sound because it gives a semantics-preserving compilation from DPDA execution over token bytes to token-level macro-transitions, and the experiments are fairly controlled through a shared SGLang setup with identical tokenizers, grammars, batching, and hardware.
* The paper has a clean systems thesis: move grammar resolution offline, represent runtime enforcement as mask[state, category] plus next_state[state, category], and thereby turn parser-style execution into regular tensor operations. The empirical section supports this thesis with cross-model scaling results, grammar-only measurements, and ablations showing that both token pre-alignment and token categorization are necessary.

Weakness:
* The gains are most pronounced at large batch sizes, while the paper itself notes that at low batch sizes the benefits are less pronounced and TTFT can offset part of the runtime advantage. This makes the contribution compelling for throughput-oriented serving, but less clearly beneficial for latency-sensitive or small-batch applications.

---

> ### Author Rebuttal · Authors · 2026-03-31
>
> Thank you for the positive assessment of our systems thesis, correctness direction, and experimental fairness. We address your two questions below.
>
> ## Q1. How would Gram2Token integrate with speculative decoding, and would the claimed runtime advantages still hold?
> Mechanistically, constrained decoding based on prefix validity checking and explicit grammar-state tracking composes naturally with draft-verify speculative decoding. In Gram2Token, runtime grammar enforcement is already reduced to category-level validity lookup and successor transition update: given the current grammar state, a token is mapped to its category, checked by the grammar mask, and then applied through the preprocessed transition semantics. Thus, in speculative decoding, draft tokens can be sampled under the same grammar mask; verification applies the same category lookup and transition update token by token; rollback restores the most recent accepted prefix via saved grammar-state checkpoints. Compared with methods that still rely on byte-level parser traversal at runtime, Gram2Token’s token-level transitions are more naturally compatible with speculative verification and rollback.
>
> To validate this, we integrated Gram2Token into SGLang’s speculative decoding interface, using EAGLE-3 as the backend, and compared speculative + Gram2Token against speculative + Pre3 on json-mode-eval. At **batch=64**, speculative + Gram2Token achieves **3198** end-to-end TPS, improving over speculative + Pre3 by **3166**; full results for other batch sizes are included in the supplementary figure. These results show that Gram2Token not only composes cleanly with speculative decoding, but also preserves its runtime advantage in speculative verification.
>
> ## Q2. Can the authors provide a clearer break-even analysis for preprocessing cost, as a function of grammar reuse frequency, batch size, and request length?
> Yes. We use exactly the same break-even definition and experimental setup as in our response to Reviewer **HTdf (Q1)**. On json-mode-eval, at batch=4/64/512 and directly measured the latency crossover point. Gram2Token overtakes Pre3 after **539/227/46** output tokens. This quantifies the amortization effect more clearly: Gram2Token is most advantageous when grammars are reusable, outputs are longer, and batch sizes are larger, whereas in cold-start, short-output, and small-batch settings, preprocessing cost can offset part of the runtime gain.

---

> > ### Author Rebuttal · Reviewer_S38h · 2026-04-03
> >
> > I think the rebuttal has addressed my main issue with the paper.

---

> > > ### Author Response · Authors · 2026-04-08
> > >
> > > Thank you for your final assessment. We are glad to have addressed your main issues. We will ensure that the clarifications and additional results provided during the rebuttal are fully integrated into the final version of the manuscript to enhance its completeness and technical depth.

---

### Official Review · Reviewer_HTdf · 2026-03-14

**Soundness:** 3
**Presentation:** 3
**Significance:** 3
**Originality:** 4
**Overall Recommendation:** 4
**Confidence:** 3

**Summary:**

The paper proposes Gram2Token, a framework for grammar-constrained decoding in LLMs that moves grammar enforcement from CPU runtime to GPU. The core idea is to pre-align a pushdown automaton (PDA) with tokenizer vocabulary via trie traversal, then group tokens into equivalence classes by grammar behavior, enabling O(1) table lookups during decoding. Experiments on JSON and SQL generation show throughput improvements over XGrammar, Formatron, and Pre3 at large batch sizes.

**Compliance With Llm Reviewing Policy:**

Affirmed.

**Final Justification:**

The rebuttal has addressed my main concern. I think this paper has reasonable quality and can be accepted.

**Key Questions For Authors:**

1. Is the performance gain compared to Pre3 significant given the penalty on the time between token? If we use the proposed approach, will it significantly hurt user experience in coding agents?

2. How much overheads there is with respect with the complexity of the grammar. For example, if we consider grammar for a standard programming language (e.g., C++, Python), will the overheads significantly increase?

**Limitations:**

See the questions I raised above.

**Strengths And Weaknesses:**

I think the high-level story is sound is very interesting. Offloading constrained decoding of context-free grammar to GPU is also a very important problem. One drawback I witnessed is whether the result itself is significant. In terms of total throughput, the perf gain in comparison with pre3 is quite limited across different batch sizes, and the time between token is significantly penalized. Note that code generation is quite latency sensitive (e.g., coding agent generates code again and again to pass some high-level test cases, not just grammar). I think the GPU implementation can be described more clearly.

---

> ### Author Rebuttal · Authors · 2026-03-31
>
> Thank you for the positive assessment. We address your four concerns below.
>
> ## Q1. Is the gain over Pre3 significant given the TTFT penalty?
>
> This trade-off is important. Table 1 already shows stable but moderate end-to-end gains over the strongest baseline Pre3, together with higher TTFT; e.g., for Qwen3-8B at batch=512, 4930 vs 4155 TPS and 972 vs 612 ms TTFT. As a rough sanity check, using the batch-level TPS/TTFT in Table 1 and assuming similar output lengths within a batch, the implied per-request crossover for Qwen3-8B is about 258/103/19 tokens at B=4/64/512. We therefore ran a cold-start end-to-end break-even experiment on json-mode-eval at batch=4/64/512 and directly measured the latency crossover point. Gram2Token overtakes Pre3 after **539/227/46** output tokens. In short, Gram2Token is most advantageous when grammar is reusable, outputs are longer, and batch size is larger.
>
> ## Q2. Will this hurt user experience in coding agents?
>
> We added a code-oriented stress test using the C grammar from WG14 N1570 Annex A, converted to the EBNF used in our implementation. Under this C-grammar setting and HumanEval-X tasks, we measured cold-start end-to-end break-even at batch=64. Gram2Token overtakes Pre3 after **145** output tokens. Since code-generation outputs are typically longer and coding agents often regenerate code under the same grammar constraints, these scenarios are more likely to cross the cold-start break-even; grammar reuse further reduces the effective preprocessing disadvantage. Thus, for grammar-constrained, reusable, and relatively long code-generation subroutines, Gram2Token is more likely to improve end-to-end latency and user experience.
>
> ## Q3. How does overhead scale with grammar complexity?
>
> We characterize grammar complexity using source-grammar indicators (maximum nesting depth and number of productions) and a state-space indicator (reachable states), and report the corresponding preprocessing cost, table memory, and steady-state TPS under batch=64. As shown in **Table 1**, complexity increases preprocessing overhead and reachable state space, but the growth is mainly paid offline rather than in per-step runtime grammar enforcement; Gram2Token still retains a steady-state advantage at larger batch sizes.
>
> **Table 1. Complexity scaling results.**
>
> **(a) JSON-mode-eval by schema depth**
> | Depth Group | n | Avg. Depth | Compile/Preprocess (ms) | Table Memory | Token Categories|
> |---|---:|---:|---:|---:|---:|
> | Depth = 0 | 6 | 0.00 | 339.67 | 0.8881 | 3.00 |
> | Depth = 1–2 | 84 | 1.42 | 377.53 | 0.8966 | 20.67 |
> | Depth = 3–4 | 10 | 3.10 | 423.29 | 0.9061 | 29.70 |
>
> **(b) JSON-mode-eval by reachable states**
> | Reachable States Bucket | n |Compile/Preprocess (ms) |Table Memory| Token Categories|
> |---|---:|---:|---:|---:|
> | [0, 16) | 8 | 334.86 | 0.8882 | 4.00 |
> | [16, 32) | 16 | 351.30 | 0.8920 | 14.56 |
> | [32, 64) | 60 | 381.00 | 0.8967 | 21.85 |
> | [64, 128) | 16 | 426.46 | 0.9075 | 29.69 |
>
> **(c) C grammar stress test**
> | Grammar | Depth | Productions | Reachable States | TTFT (ms) | Table Memory (MB) | Steady-state TPS |
> |---|---:|---:|---:|---:|---:|---:|
> | C grammar (full raw Annex A.2) | 12 | 370 | 4116 | 1758.39 | 1.0804 | 1998 |
> | C grammar (minimal subset) | 11 | 93 | 173 | 1254.12 | 0.1650 | 1902 |
>
> ## Q4. I think the GPU implementation can be described more clearly.
>
> Thank you for this suggestion. We agree that the runtime GPU path can be described more directly. Gram2Token no longer performs explicit byte-level parser traversal or reduction-path exploration at runtime. Instead, grammar effects are preprocessed into token-level transition semantics, and runtime only performs lightweight category lookup, validity masking, and the corresponding state/stack update. This is consistent with Section 3.4, where runtime grammar enforcement reduces to category-level mask lookup and successor transition lookup.
> The runtime flow can be summarized as follows:
> ```text
> Given current grammar state s_t and candidate token v:
> 1. k <- cat(v)
> 2. valid <- M_tilde[s_t, k]
> 3. if valid:
>        (s_{t+1}, stack_update) <- T_tilde(s_t, k)
>        apply lightweight state/stack update
>    else:
>        mask out v
> ```

---

> > ### Author Rebuttal · Reviewer_HTdf · 2026-04-03
> >
> > I think the rebuttal has addressed my main issue with the paper.

---

> > > ### Author Response · Authors · 2026-04-08
> > >
> > > Thank you for your positive final feedback and for recognizing the effectiveness of our rebuttal. We are pleased that our responses and new experimental data have successfully addressed your concerns.
> > >
> > > We are committed to delivering a high-quality final revision that incorporates the full breadth of our discussion.

---

### Decision · Program_Chairs · 2026-04-30

**Decision:**

Accept (regular)

**Comment:**

This paper proposes Gram2Token, a GPU-native approach to grammar-constrained decoding that compiles grammar logic into token-level lookup tables. The authors appear to consider an important context: scalable structured generation in LLM systems. A pressing challenge examined by this paper is eliminating CPU–GPU synchronization overhead while preserving exact grammar correctness.

Reviewers agree the method is technically sound, well-motivated, and demonstrates clear throughput gains, especially at large batch sizes. The key insight—shifting grammar enforcement offline and reducing runtime to O(1) GPU lookups—is seen as clean and impactful.

Concerns focus on higher preprocessing cost (TTFT), limited benefits in low-latency or small-batch settings, and scalability to complex or heterogeneous grammars. The rebuttal addressed these issues with additional experiments and clarifications, resolving most concerns.

Overall, this is a solid systems contribution with practical relevance, particularly for throughput-oriented LLM serving, and meets the bar for acceptance.